# Regulation of Long Non-Coding RNAs by Plant Secondary Metabolites: A Novel Anticancer Therapeutic Approach

**DOI:** 10.3390/cancers13061274

**Published:** 2021-03-13

**Authors:** Mohammad Reza Kalhori, Hamid Khodayari, Saeed Khodayari, Miko Vesovic, Gloria Jackson, Mohammad Hosein Farzaei, Anupam Bishayee

**Affiliations:** 1Medical Biology Research Center, Health Technology Institute, Kermanshah University of Medical Sciences, Kermanshah 6714415185, Iran; mohammadreza.kalhori@kums.ac.ir; 2International Center for Personalized Medicine, 40235 Düsseldorf, Germany; h.khodayari@hotmail.com (H.K.); saeed.khodayari@hotmail.com (S.K.); 3Breast Disease Research Center, Tehran University of Medical Sciences, Tehran 1419733141, Iran; 4Department of Mathematics, Statistics, and Computer Science, University of Illinois at Chicago, Chicago, IL 60607, USA; miko@math.uic.edu; 5Lake Erie College of Osteopathic Medicine, Bradenton, FL 34211, USA; gloriajackson229@gmail.com; 6Medical Technology Research Center, Health Technology Institute, Kermanshah University of Medical Sciences, Kermanshah 6718874414, Iran

**Keywords:** lncRNAs, natural products, phytochemicals, cancer treatment, targeted therapy, precision medicine

## Abstract

**Simple Summary:**

Cancer is caused by the rapid and uncontrolled growth of cells that eventually lead to tumor formation. Genetic and epigenetic alterations are among the most critical factors in the onset of carcinoma. Phytochemicals are a group of natural compounds that play an essential role in cancer prevention and treatment. Long non-coding RNAs (lncRNAs) are potential therapeutic targets of bioactive phytochemicals, and these compounds could regulate the expression of lncRNAs directly and indirectly. Here, we critically evaluate in vitro and in vivo anticancer effects of phytochemicals in numerous human cancers via regulation of lncRNA expression and their downstream target genes.

**Abstract:**

Long non-coding RNAs (lncRNAs) are a class of non-coding RNAs that play an essential role in various cellular activities, such as differentiation, proliferation, and apoptosis. Dysregulation of lncRNAs serves a fundamental role in the progression and initiation of various diseases, including cancer. Precision medicine is a suitable and optimal treatment method for cancer so that based on each patient’s genetic content, a specific treatment or drug is prescribed. The rapid advancement of science and technology in recent years has led to many successes in this particular treatment. Phytochemicals are a group of natural compounds extracted from fruits, vegetables, and plants. Through the downregulation of oncogenic lncRNAs or upregulation of tumor suppressor lncRNAs, these bioactive compounds can inhibit metastasis, proliferation, invasion, migration, and cancer cells. These natural products can be a novel and alternative strategy for cancer treatment and improve tumor cells’ sensitivity to standard adjuvant therapies. This review will discuss the antineoplastic effects of bioactive plant secondary metabolites (phytochemicals) via regulation of expression of lncRNAs in various human cancers and their potential for the treatment and prevention of human cancers.

## 1. Introduction

Cancer is caused by the rapid and uncontrolled growth of cells that eventually lead to tumor formation. Failure of early diagnosis and treatment can lead to metastasis or death. Any event that increases the expression of oncogenes or decreases the expression of tumor suppressor genes can result in cancer [1]. Genetic and epigenetic alterations are among the most critical factors in the onset of carcinoma, due to the altered expression of gene-coding and non-coding RNAs (ncRNA) that regulate apoptosis, proliferation, and differentiation [2,3]. Nevertheless, several lifestyle factors, such as physical inactivity, alcohol consumption, smoking, poor nutrition, and exposure to ultraviolet radiation or X-rays, can contribute to healthy cells’ transition into malignant cells [4]. Surgery, chemotherapy, and radiotherapy are standard and conventional methods of cancer treatment. However, due to chemotherapy and radiotherapy’s side effects and resistance to treatment in some patients, many researchers have turned their attention to alternate approaches, including natural compounds [5].

Precision medicine is a suitable and individualized treatment method for each patient, followed by molecular diagnostic tests of disease at the molecular level based on the genetic content, phenotypic, biomarker, or psychosocial characteristics of each patient [6]. This method allows the medicine to be prescribed only for patients who benefit from that and reduce the failure rate of pharmaceutical clinical trials, also saves costs and avoids side effects in individuals who do not benefit from that treatment [7]. Precision medicine is an approach for preventing and treating various diseases, including cancer, that considers the patient’s lifestyle, environment, and genetic diversity for treatment. Therefore, treating a patient with a particular cancer is different from other people who have the same type and stage of cancer [8].

Of the total human genome, only 2% is transcribed into proteins, with the remaining regions encoded as non-coding RNAs [3]. Long non-coding RNAs (lncRNAs) are a class of non-coding RNAs containing more than 200 nucleotides in length, which play an essential role in cell biological activities, such as differentiation, proliferation, apoptosis, and growth. Dysregulation of lncRNAs plays a fundamental role in developing various diseases, such as cancer [9]. Most lncRNAs are localized and function within the nucleus, but some only operate in the cytoplasm. lncRNAs can stimulate or suppress transcription by various mechanisms, including epigenetic changes (such as chromatin remodeling or histone modification), transcription factor decay, microRNA sponge, a scaffold for various proteins, and splicing regulation [10]. Altered expression of lncRNAs, such as PTENP1, linc-PINT, and GAS5, plays an essential role in regulating apoptosis in cancer cells [11,12,13]. In human embryonic and adult life, angiogenesis plays a vital role in several physiological processes. Nevertheless, ectopic angiogenic processes have also been associated with the pathogenesis of cancer. Since tumor cells need more nutrients, oxygen, and the removal of metabolites than normal cells, this leads to angiogenesis activation [14]. Moreover, one of the preconditions for cancer metastasis is epithelial-to-mesenchymal transition (EMT) [15]. lncRNAs are essential regulators of tumor angiogenesis and EMT in various cancers such as gastrointestinal, lung, breast, and brain tumors by regulating oncogenic pathways, angiogenic factors, and epigenetic alterations [16]. Therefore, lncRNAs, such as MALAT1, UBE2CP3, HULC, UCA1, and PCA3 play an essential role in cancer metastasis by altering gene expression [17,18,19]. 

Phytochemicals are a group of natural compounds extracted from fruits, vegetables, and other plants that can play an essential role in cancer prevention and treatment due to their anti-inflammatory, antioxidant, and anticancer properties [20]. One reason for the failure of cancer treatment is the resistance of cancer cells to chemotherapy, and the other is that anticancer drugs are unable to differentiate normal proliferating cells from malignant cells [21]. Genetic and epigenetic modifications in cancer cells cause the expression of surface and intracellular proteins to be different from normal cells [22]. From ancient times, natural compounds could improve many diseases, including cancer, without any side effects. Previous studies have shown that lncRNAs are potential therapeutic targets of bioactive phytochemicals. These compounds could regulate the expression of lncRNA directly and indirectly without any side effects [23,24]. In vivo and in vitro studies reveal that phytochemicals could inhibit proliferation, invasion, migration, EMT, metastasis and induce chemosensitization and radiosensitization of cancer cells by downregulation of oncogenic lncRNAs or upregulation of tumor suppressor lncRNAs [25,26,27]. 

To date, only four review articles [28,29,30,31] evaluated the association between lncRNAs and the overexpression of various tumor-associated proteins and the role of phytochemicals on the expression of lncRNAs in cancers and chronic human diseases, either indirectly or directly by affecting a wide range of upstream molecules. Moreover, in these articles, the in-depth mechanisms by which phytochemicals modulate cancer-related lncRNAs are lacking [28,29,30,31]. Hence, this study aims to analyze the latest published articles systematically in this field for the first time. The present review tries to decipher the role of lncRNAs in the progression and development of human cancer and its regulation by various phytochemicals in the medicinal plant according to the recent advances and clinical significance in this field. The following sections will discuss in vivo and in vitro anticancer effects of phytochemicals in numerous human cancers via regulation of lncRNA expression and their downstream target genes.

## 2. Methodology for Literature Search and Selection

We have used Preferred Reporting Items for Systemic Reviews and Meta-Analysis (PRISMA), a reliable method for compiling systematic reviews [32] for this work. A systematic search was performed on Scopus, PubMed, Science Direct, Google Scholar and Cochrane Library electronic databases to find articles using the following combination of terms: “cancer, malignancy OR neoplasms” AND “lncRNA” AND “phytochemical, flavonoid, polyphenols, herbal, natural agents, plants, natural products, anacardic acid, apiin, baicalein, berberine, bharangin, calycosin, curcumin, kaempferol glycosides, glycyrrhizin, naringin, juglanin, baicalin, luteoloside, tiliroside, reserpine, hesperidin, kaempferol, triglycoside, fisetin, naringenin, isoflavone, hesperetin, pinostrobin, rhamnetol, routine, rhamnazol, apigenin, rhamnocitro, flavanone, flavonol, epigallocatechin, gambogic, genistein, ginsenoside, hyperoside, luteolin, polydatin, quercetin, resveratrol, sanguinarine, silibinin, and sulforaphane, emodin, silymarin, eohesperidin, gambogic acid, daidzein, nobiletin, ribavirin, sinomenine, taxol, cryptotanshinone, puerarin, echinacoside, oxymatrine, triptolide, ursolic acid, and osthole”. The last search was conducted in November 2020. Two authors independently screened the results (1181 articles) and identified related articles.

Articles that did not report the role of phytochemicals in cancer inhibition using lncRNAs were excluded. Finally, 60 original articles about the role of phytochemicals in the regulation of lncRNAs expression in human cancers were filtered and included in this study. Of these, six (10%), two (3.3%), five (8.3%), four (6.6%), three (5%), fourteen (23.3%), five (8.3%), five (8.3%), two (3.3%) and three (5%) articles were related to ginsenoside, quercetin, baicalein, resveratrol, berberine, curcumin, genistein, and epigallocatechin, gambogic acid and calycosin, respectively. Furthermore, 16.6% of all articles were related to anacardic acid, bharangin, hyperoside, luteolin, polydatin, quercetin, sanguinarine, silibinin, sinomenine, and sulforaphane. The process of literature searches and study selection is depicted in Figure 1.

## 3. Non-Coding RNAs

Non-coding RNAs (ncRNAs) constitute a group of RNAs that are not converted by ribosomes into proteins. The ncRNAs are divided into two classes: regulatory ncRNAs and structural ncRNAs [19]. The regulatory ncRNAs include small ncRNAs (20–50 nucleotides), medium ncRNAs (50–200 nucleotides), and long non-coding RNAs (lncRNAs) (longer than 200 nucleotides). Among essential structural ncRNAs are ribosomal RNAs (rRNAs) and transfer RNAs (tRNAs) [33]. 

### 3.1. Categories of lncRNAs

Previous studies have confirmed that lncRNAs are the most significant ncRNAs, playing an essential role in the advancement and development of malignant and non-malignant human diseases. Cell stages and/or tissue types contribute to lncRNAs function and biogenesis mechanisms. They are also subdivided into several groups depending on their biogenesis, function, and structure [33,34]. lncRNAs are referred to as antisense, bidirectional, enhancer, intergenic, and intronic based on their origins [35,36]. Similar to messenger RNAs (mRNAs), some lncRNAs appear to be capped at their 5’ end and polyadenylated at their 3’ end, respectively, following RNA-polymerase II-mediated transcription. Biogenesis of lncRNAs can be provided by several agents, including ribonuclease P (RNase P)-mediated cleavage for the generation of mature ends, the formation of small nucleolar RNA (snoRNA), and small nucleolar ribonucleoprotein (snoRNP) complex caps at their ends, and formation of linear and circular lncRNAs [33,37].

### 3.2. Functions of lncRNAs

In humans, the lncRNAs are functionally involved in all processes of normal and malignant cell development and differentiation [38]. In multiple-factor cross-linking, lncRNAs have been shown to alter gene expression by inducing epigenetic modifications, such as chromosome remodeling, histone modification, and nucleosome positioning changes. Another principal mechanism of lncRNAs is chromatin remodeling regulation through interaction with switching defective/sucrose nonfermenting (SWI/SNF) and chromatin remodeling complex [39,40]. As a chromosome reconstitution complex, SWI/SNF comprises multiple subunits and plays a crucial role in nucleosome positioning changes [41]. lncRNAs also contribute to gene expression regulation by adjusting histone modifications, including acetylation, de-methylation, and histone methylation. lncRNAs mostly target S-adenosyl-L-homocysteine hydrolase, DNA damage-inducible protein α (GADD45A), Sirtuin6 (SIRT6), and polycomb repressive complex 2 (PRC2) [42,43,44]. They also have a regulatory effect on the expression of nearby coding genes, including transcription factors such as tumor protein 53 (TP53 or p 53), nuclear factor-κB (NF-κB), octamer-binding transcription factor 4 (Oct4), sex-determining region Y-box2 (Sox2), and c-Myc proto-oncogene protein [45,46]. Many of these factors mediate several signaling pathways and play a key role in migration, apoptosis, differentiation, and proliferation. The Tumor-suppressor p53 gene helps regulate various genes involved in apoptosis, cell cycle, and DNA repair. The expression of p53 protein is typically balanced by translation, post-translational modifications (PTMs), and protein stability [47]. Numerous studies have described the significance of lncRNAs in the p53 gene regulatory network. Furthermore, lncRNAs are also known to mediate the NF-κB signaling pathway. lncRNAs can also regulate the Akt (also referred to as protein kinase B) pathway, Notch, and canonical Wnt pathway via a cross-linking network [48,49].

## 4. Role of lncRNAs in Human Cancers

Carcinogenesis (tumorigenesis or oncogenesis) is the process that leads to the formation of cancer cells [50]. As such, carcinogenesis involves the following steps: (1) the emergence of mutations in single or multiple DNA repair and tumor-suppressor genes and (2) the formation of oncogenes mediated by a mutation in proto-oncogenes. These eventually develop specific transformed malignant cells with some unique characteristics, known as hallmarks of cancer [51]. Eight hallmarks of cancer have been identified, categorized into three distinct classes: abnormal growth, increased motility, and altered mode of energy metabolism [51]. The first class includes growth suppressor evasion, immortality, immune destruction evasion, and proliferative signaling sustainability. The second class includes angiogenesis induction, invasion initiation, and metastasis. Finally, the third class includes energy metabolism reprogramming [51]. Many studies have demonstrated the significance of lncRNAs in carcinogenesis and their specific effect on cancer hallmarks (Figure 2). 

In cancer, lncRNAs can serve as a tumor-suppressive agent, an oncogenic agent, or both [17]. A large number of lncRNAs are linked with each hallmark of cancer. Among the well-known lncRNAs involved in tumor progression in humans, those associated with proliferation, survival and migration of cancer cells are presented in Table 1. This table attempts to describe detailed information of each lncRNA, including oncogenic or tumor-suppressive activity, associated cancer, molecular function, and outcome. 

### 4.1. Role in EMT

It has been shown that these ncRNAs mainly target some of the essential signaling pathways and transcription factors in cancer cells. Additionally, lncRNAs can influence the metabolism of cancer cells [57]. Shi et al. [69] identified that MALAT1, a prominent oncogenic lncRNA in human cancers, induces EMT. Induction of EMT destroys the polarity of epithelial cells and reduces intercellular adhesion. The MALAT1/miR-124/calpain small subunit 1 (capn4) axis has been demonstrated to be one of the key factors contributing in EMT, invasion, and proliferation of nasopharyngeal carcinoma (NPC) cells [69]. Furthermore, MALAT1 regulates tumor angiogenesis and progression by activating phosphatidylinositol 3-kinase (PI3K)/Akt, extracellular signal-regulated kinase (ERK)/mitogen-activated protein kinase (MAPK), and Wnt/β-catenin signaling pathways [71]. Like MALAT1, ANRIL also contributes to the induction of EMT. The ANRIL, as an oncogenic lncRNA, can facilitate the survival and proliferation of malignant cells. ANRIL has been shown to inhibit apoptosis through the silencing of Kruppel-like factor 2 (KLF2) and p21 genes [54]. It can also facilitate cancer metastasis and cell invasion by introducing matrix metalloproteinase-3 (MMP-3 or stromelysin-1) protein expression [53,55]. Moreover, lncRNAs, such as CASC9, CCAT1, ZEB2-AS1, and lncTCF7, are involved in EMT with different molecular mechanisms, and regulation of their expression can play a therapeutic role for cancer [81,82].

### 4.2. Role in NF-κB Signaling and Telomerase Activity

Nuclear factor-κB (NF-κB) helps active cancer cells’ responses to the inflammatory microenvironment and regulates the migration, proliferation, and survival of neoplastic cells [83]. GAS5, a lncRNA tumor suppressor, targets NF-κB. GAS5 expression can prevent the activation of NF-κB and ERK1/2 pathways. It can also control the cell cycle by arresting cells in the G0 phase [13]. HULC is a lncRNA that is upregulated in lung squamous cell carcinoma. This lncRNA has a negative effect on protein tyrosine phosphatase receptor type O (PTPRO) via phosphorylation and activation of NF-κB and enhances the proliferation of LSCC cells [84]. NF-κB interacting lncRNA (NKILA) works as a tumor suppressor lncRNA, and its low expression in laryngeal cancer is related to shorter overall survival. Overexpression of NKILA can promote apoptosis and sensitize laryngeal cancer cells to radiation via repression of IκB phosphorylation and NF-κB activation [85]. Additionally, HOTAIR by negative regulation of miR-218 in CRC cells can activate NF-κB signaling and promotes 5-FU resistance and proliferation [86].

Additionally, lncRNAs play an essential role in maintaining cancer cell immortality by regulating telomerase function. Malignant cells have been shown to avoid apoptosis by using telomerase to increase telomeres’ size by inserting repeats onto the edge 3’ of chromosomes [87]. Telomeric repeat-containing RNA (TERRA) includes a heterogeneous class of telomeric region-transcribed lncRNAs [76]. As a lncRNA tumor suppressor, the expression of TERRA contributes to the negative regulation of telomerase activity, whose downregulation plays a key role in carcinogenesis. RB and TP53 genes have been found to regulate TERRA expression [76,78]. Tan et al. [88] show that MALAT1 by regulation of TERT can increase the telomerase activity in bone marrow mesenchymal stem cells. lncRNA MEG3 has been downregulated in non-small cell lung cancer and has a close relation with poor prognosis in patients. Its overexpression in A549 cells could inhibit invasion, migration, proliferation, and telomerase activity by downregulating Dyskeratosis congenita 1 (DKC1) [89]. DKC1 is an essential subunit of ribonucleoprotein telomerase which plays an important role in the stabilization and activity of telomerase [90].

### 4.3. Role in Energy Metabolism

Unbalanced metabolic homeostasis by high energy production induces tumor growth, cancer progression, and metastasis. Glucose metabolism is involved in glucose uptake, lactate production, and oxidative phosphorylation. Uncontrolled growth, survival and drug resistance of cancer cells are linked to elevated glucose metabolism [91]. In cancer cells, a transformed mode of energy metabolism occurs in response to tumor oxygenation, which may affect other important aspects of malignancies regulated by lncRNAs. Hypoxia-inducible factor-1α (HIF-1α) has proved effective in facilitating the expression of specific lncRNAs, such as in UCA1, under hypoxic conditions [92]. Xu et al. [93] showed that UCA1, as an oncogenic lncRNA, assists in the regulation of the proliferation, invasion, and metastasis of cancer cells by targeting the KLF832 transcription factor and activating MMP14, fibroblast growth factor receptor-1 (FGFR-1)/ERK, and zinc finger E-box binding homeobox 1 and 2 (ZEB1/2)-fascin homolog 1 (FSCN1) pathways [19,79,80]. Furthermore, LINK-A lncRNA expression has been identified as an essential factor in facilitating the glycolytic reprogramming and tumorigenesis of cancer cells by triggering signaling pathways dependent on LINK-A, e.g., Akt, leading to carcinogenesis and the development of resistance to Akt inhibitors [48]. Previous studies reveal that lncRNA-p23154 and MACC1-AS1 are upregulated in human cancers, and these lncRNAs could promote cellular glycolysis and glucose metabolism via modulation of glucose transporter 1 (GLUT1) [94,95].

### 4.4. Role in Drug Resistance

Recently, remarkable evidence has been obtained regarding the regulatory roles of lncRNAs in therapeutic responses of cancer cells to chemotherapy drugs. The term of drug resistance is reoffered to an inherent character of malignant cells to administrated antitumor agents, ending in poor overall survival of cancer patients [96]. In Table 1, some of the well-known lncRNAs involved in the event of cancer drug resistance have been summarized. The principal mechanisms of drug resistance in cancer cells are: (1) overexpression of drug efflux transporters like ATP-binding cassette (ABC) superfamily pumps [97]; (2) enhanced repair of chemotherapy induced-genome damages [98]; (3) alternation signaling pathways controlling drug resistance [99]; (4) enhanced modulation of drug metabolism; (5) increasing resistance to cell death mechanisms [100]; and (6) EMT [99]. 

ANRIL is one of the well-studied lncRNAs inducing chemotherapy resistances in cancer cells. Through an experimental study, Li and Zhu, (2019) have reported a tight correlation of ANRIL with cisplatin resistance in osteosarcoma cells. In ANRIL-depleted cancer cells, they have observed a significant improvement of cisplatin sensitivity via miR-125a-5p/STAT3 pathway [101]. Furthermore, ANRIL has a profound impact on paclitaxel sole resistance. It has been demonstrated that ANRIL overexpression could powerfully inhibit the apoptotic death of lung adenocarcinoma cells following the paclitaxel sole treatment [102]. Likewise, the expression of lncRNA HOTAIR in lung tumors has been linked to chemoresistance. Within the downregulation of p21 and upregulation of Kruppel-like factor 4 (KLF4), it has been found that HOTAIR overexpression could influence the cisplatin resistance in the lung tumors [103,104]. Through an experimental study, Guo et al. (2018) observed a significant improvement of cisplatin sensitivity in HOTAIR-depleted lung cancer cells following the inhibition of multidrug resistance-associated protein-1 (MRP-1) and Wnt signaling pathway [105]. Docetaxel, doxorubicin, gemcitabine, gefitinib, methotrexate, oxaliplatin, sunitinib, tamoxifen, trastuzumab, and 5-FU are the other types of anticancer agents with antitumor efficacies negatively regulated by lncRNAs [106]. The evidence of current studies has clearly shown the interfering effects of lncRNAs on the outcomes of conventional treatments of cancer, such as chemotherapy. Simultaneous administration of complementary agents targeting lncRNAs in tumors could be an effective approach for improving the efficiency of chemotherapies.

## 5. Plant lncRNAs 

Plants, like other eukaryotes, can regulate their gene expression using lncRNA. The expression levels of lncRNAs are different in various tissues and can change in response to environmental stresses [107]. Several databases, including PNRD, Plant NATsDB, and EVLncRNA, are available to study plant lncRNAs. The EVLncRNA database collected and presented approximately 4010 lncRNAs in 124 species [108]. ENOD40 is the first plant lncRNA that was isolated from Medicago plants. This lncRNA could regulate symbiotic nodule organogenesis in *Medicago truncatula* [109]. lncRNA in plants plays a vital role in regulating the flowering time, modulation of reproductive organ development, seedling photomorphogenesis, root organogenesis, response to biotic and abiotic stresses by influencing different biological processes and molecular mechanisms [110,111]. For example, lncRNA973 can increase salt stress resistance in cotton (*Gossypium* spp.) by regulating a series of salt stress-related genes [111]. Unlike mammals, plants have RNA polymerases of Pol IV and Pol V [112]; they have lncRNA, which is not found in mammals. Therefore, it is necessary to investigate whether the treatment of cancer cells with plant lncRNAs could increase or inhibit cell proliferation. It would be interesting to find out if consumption of plants by laboratory animals causes detectable amounts of plant lncRNAs in their blood and can affect tumor growth. Thus far, most human knowledge about plant’s lncRNAs has been based on their role in regulating plant biological processes, and no study has been published to establish that plants’ lncRNAs can be useful in the treatment of human cancer.

## 6. Phytochemicals as Antitumor Agents by Targeting lncRNAs

Natural products or their semi-synthetic derivatives have been used in clinics for decades as anticancer chemotherapeutic drugs [113]. Secondary plant metabolites (known as phytochemicals) are a group of natural products obtained from roots, fruits, leaves, and shoots of plants. Phytochemicals by several biological features such as antioxidant and anticancer activity are responsible for health-promoting and disease prevention [114]. According to chemical structures, phytochemicals are classified into four main groups of phenolics and polyphenolics (e.g., quercetin, luteolin, genistein, resveratrol, calycosin, daidzein, and genistein), nitrogen-containing alkaloids (e.g., berberine and sanguinarine), terpenes (e.g., betulin), and sulfur-containing compounds (e.g., sulforaphane) [113]. Phytochemicals regulate oncogenic transcription factors, tumor suppressor signaling pathways, and cytogenetic processes to exert their proliferative and tumor inhibitory effects [115]. The following sections present various phytochemicals that exhibit antineoplastic effects by altering the expression of lncRNAs in various cancer models (Table 2).

### 6.1. Anacardic Acids

Anacardic acids are phenolic lipids that are plentiful in the shell of the cashew nut (*Anacardium occidentale*). A specific congener of anacardic acid (AnAc 24:1n5) could downregulate the expression of endogenous estrogen regulatory genes, such as CTSD, CCND1, and TFF1, in breast cancer cell lines [163]. Schultz et al. [116] revealed that this phytochemical could increase PDK4 and decrease the expression of TGM2, INSIG1, and SCD in MCF-7 and MDA-MB-231 breast cancer cell lines by altering the expression of a series of lncRNAs.

### 6.2. Baicalein

Baicalein is a flavonoid compound extracted from the root of *Scutellaria baicalensis* Georgi, which has various pharmacological properties that allow its use as an adjunct therapy against various diseases, including cancer [164]. One of the factors involved in cancer development and pathophysiology is the abnormal expression of lncRNAs in tumors compared to non-tumor tissues. For example, colon cancer-associated transcript 1 (CCAT1) is a lncRNA reported to increase the incidence of melanoma cancer [165]. Baicalein exerted its anticancer properties in melanoma cancer cells (IC_50_: 50 µM, 24 h) by inhibiting CCAT1 expression. In addition to reducing cell proliferation by increasing the percentage of apoptotic cells, this natural compound could reduce cell invasion and migration. Moreover, inhibiting CCAT1 expression inhibited the MEK/ERK and Wnt/β-catenin pathways axis by reducing the expression of Wnt-3a, β-catenin, MEK, and ERK genes [117]. Baicalein could reduce tumor growth, migration, and proliferation in cervical cancer xenograft tumors and cervical cell lines in both a time and concentration-dependent manner. Thus, it may be useful in the clinical treatment of cervical cancer (Table 3). Further investigations have revealed that the anticancer action of baicalein occurs via the downregulation of lncRNA-BDLNR. Therefore, the ectopic expression of BDLNR can prevent the anticancer effect of baicalein in cervical cancer [23].

Baicalein can also regulate the expression of antioncogenic lncRNA in breast cancer and hepatocellular carcinoma (HCC) [118,119]. lncRNA-NKILA is a negative regulator of NF-κB activity and is downregulated in HCC. Baicalein may reduce inhibitors-of-κBα (IκBα) phosphorylation and NF-κB activity by increasing NF-κB-interacting lncRNA (NKILA) expression, thereby inhibiting migration, proliferation, and inducing apoptosis [118]. PAX8-AS1-N is a lncRNA that is downregulated in breast cancer tumors. Low expression of PAX8-AS1-N in patients with these tumors is related to poor survival, and reducing its expression with siRNAs in breast cancer cell lines leads to the induction of cell proliferation and reduced apoptosis. Treatment of breast cell lines with baicalein can significantly increase this lncRNA. Therefore, PAX8-AS1-N could induce tumor suppressor proteins (ZBTB4, PTEN, and CDKN1A) by inhibiting miR-17-5p, resulting in the induction of cell death and repression of cell proliferation [119].

### 6.3. Berberine

Berberine is an anticancer isoquinoline alkaloid found in Berberis plants, such as *Coptis chinensis* [166]. CASC2 is an anticancer lncRNA that is downregulated in many cancers, including colorectal cancer [167]. Results have shown that in a time- and concentration-dependent manner (IC_50_: 40 µM, 48 h), berberine was able to induce apoptosis, and inhibit proliferation and viability in CRCs by increasing CASC2 expression. In fact, CASC2 performs its tumor suppressor function by interaction with AUF1 to inhibit Bcl-2 mRNA translation and to inhibit CASC2 expression by siRNA, blocking berberine’s anticancer effect [26,120]. HOTAIR is an oncogenic lncRNA that is located on chromosome 12 inside the Homeobox C (HOXC) gene cluster and plays an essential role in carcinogenesis and tumorigenesis; its upregulation was first reported in breast cancer [168]. 

Berberine also has the potential to be used as a chemosensitizer in cancer treatment [169]. Due to its synergistic effect on gefitinib, berberine (25 µmol/L) significantly reduces proliferation, invasion, and metastasis compared to the gefitinib (5 µmol/L) group alone in non-small-cell lung carcinoma (NSCLC). Additionally, berberine can increase miR-34a-5p expression by downregulation of HOTAIR because HOTAIR functions as a sponge molecule for miR-34a [170]. Consequently, berberine indirectly reduces vimentin, and increases E-cadherin via miR-34a-5p upregulation. Interestingly, the ectopic expression of HOTAIR will eliminate the effect of berberine on EMT [27]. Considering berberine can reduce HOTAIR expression in vitro and in vivo, it has the potential to be an appropriate treatment for cancers such as NSCLC, where this lncRNA is increased.

### 6.4. Bharangin 

Bharangin is a phytochemical obtained from the root nodules of a medicinal plant, *Pigmacopremna herbacea* [171]. The previous study showed the potential anticancer effect of bharangin against MDA-MB-231, MCF-7, MDA-MB-453, and MDA-MB-468 breast cancer cell lines. Bharangin via regulation H19, MEG-3, GAS-5, MHRT, NEAT1 lncRNAs, and suppression NF-κB activity could inhibit migration, proliferation, and cell cycle of breast cancer cell lines [121]. Bharangin also sensitize tumor cells to chemotherapeutic agents and induce apoptosis in lymphoma, multiple myeloma, and leukemia cells [171].

### 6.5. Calycosin 

Phytoestrogens (dietary estrogens) are a group of non-steroidal plant compounds found in plants that structurally are very similar to human estrogen. Plant-derived phytoestrogens are divided into four main categories: isoflavones, lignans, coumestans, and stilbenes [123]. Isoflavones such as calycosin and genistein have anticancer activity and can reduce the risk of estrogen-related cancers [172]. Calycosin (C16H12O5), the main active ingredient extracted from *Radix astragali*, is widely used as a traditional Chinese herbal medicine [124]. Ewing sarcoma-associated transcript 1 (EWSAT1) is a lncRNA located on chromosome 15 that is upregulated in cell lines and tissues of NPC. In vitro and in vivo studies on NPC cells have shown that calycosin can reduce tumor growth and cell proliferation by decreasing the expression of EWSAT1 and its downstream genes, including p-TAK1, TRAF6, and p-IκBα [124]. Consumption of higher levels of phytoestrogens is likely to be one reason why breast cancer incidence in East Asian countries is lower compared to developed European countries and the United States [173]. To confirm this hypothesis, Tian et al. [122] investigated the treatment of both ER+ (T47D and MCF-7) and ER− (SKBR3 and MDA-MB-468) breast cancer cells with calycosin. In vitro (16 μM) and in vivo (55 mg/kg) studies reveal that calycosin inhibits the growth of breast cancer cells in a concentration-dependent manner through upregulation of WDR7-7-GPR30 signaling. Furthermore, the inhibitory effects of calycosin on ER+ cells were more significant than ER-breast cancer cells. This effect may be because calycosin could downregulate ERα and miR-375 and could upregulate RASD1 expression in ER+ cells [122].

### 6.6. Curcumin

Curcumin is the main natural polyphenol in the rhizome of *Curcuma longa* plants, and has been reported to have many anticancer effects [174,175]. This compound, without any side effects, could induce apoptosis and enhance chemoradiation sensitivity [176]. One of the challenges of cancer treatment is resistance to chemotherapy due to cancer stem cells present. Curcumin plays an essential role in inducing hypersensitivity to chemotherapy in the gemcitabine-resistant pancreatic ductal adenocarcinoma (PDAC) cell by inhibiting the PRC2/PVT1/c-Myc axis [125]. In vivo and in vitro studies reveal curcumin has a cytotoxic effect in PDAC cells and has a synergistic effect on gemcitabine, and improves the sensitivity of BxPC3-GemR cells to gemcitabine via downregulation of lncRNA-PVT1 and enhancer of zeste homolog 2 (EZH2) [125]. Over the past decade, many studies have shown that curcumin plays a significant role in the radiosensitivity of different cancer models [177]. In NPC cells, curcumin could downregulate growth factor receptor-bound protein 2 (GRB2), signal transducer and activator of transcription 3 (STAT3), epidermal growth factor receptor (EGFR), and enhance radiosensitivity by regulating the circRNA-miRNA pathway [178]. Wang et al. [179] showed that curcumin could also increase NPC cells’ radiosensitivity by modifying the expression of 116 lncRNAs that are dysregulated by radiation.

Deregulation of *linc*-*PINT* plays an essential role in the pathogenesis of various cancer types, including acute lymphoblastic leukemia [126,180]. The use of compounds that upregulate its expression may play a vital role in improving this cancer. Curcumin could upregulate this lncRNA in the MOLT4 (human T lymphoblast; acute lymphoblastic leukemia) cell line to induced cell cycle arrest and apoptosis by increasing the level of HOMX1 [126]. The p53 is a critical tumor suppressor protein that plays an essential role in cancer prevention by induction of cell cycle arrest, apoptosis, and controlled cell proliferation [181]. lncRNA-H19 and lncRNA-ROR by inhibiting p53 perform a fundamental role in tumor pathogenesis and have the ability to be inhibited by curcumin [24,182]. In vitro studies on gastric cancer cells have shown that curcumin via an effect on c-Myc/H19/p53 axis downregulated c-Myc and lncRNA-H19, thereby altering the expression of Bcl-2, p53, and Bax proteins, thus inducing apoptosis. Interestingly, the ectopic expression of H19 could repress the anticancer effect of curcumin [24]. 

DNA topoisomerase IIa (TOP2A) expression is associated with cell proliferation and is known to be overexpressed in many malignancies. The study by Kujundzić et al. [183] showed that low curcumin concentrations reduced H19 expression and viability by decreasing TOP2A in different tumor cell lines. The miR-145 has a binding site in the lncRNA-ROR and oct4 mRNA sequences, reducing their expression and function through the competing endogenous RNA effect. By overexpression of miR-145 and downregulation of lncRNA-ROR, curcumin could suppress the invasion, proliferation, xenograft growth, and cell cycle (G2/M) of human prostate cancer stem cells [127]. Liu et al. [184] revealed that curcumin by downregulating DNMT1 and DNMT3B could decrease the miR-145 promoter’s methylation, and thereby upregulate the expression of miR-145.

Renal cell carcinoma (RCC) is the most common form of kidney cancer; metastasis and invasion are the most critical factors in the treatment failure of this cancer [185]. Curcumin can inhibit the migration, invasion, and proliferation of RCC by downregulation HOTAIR and upregulation XIST [128,129]. XIST expression is decreased in RCC tissues and cell lines. This lncRNA can upregulate p21 by inhibiting miR-106b, therefore playing an essential role in regulating the biological activity of RCC [129]. XIST has a putative binding site in miR-106b-5p and can directly bind to miR-106b-5p. Therefore, XIST can work as a sponge molecule for miR-106b-5p [129]. Curcumin can exert its anticancer function in RCC through XIST/miR-106/p21 axis. Activation of the AMPK pathway could suppress mTOR signaling, thereby inhibiting colony formation, cell proliferation, and cell cycle in CRCs. A study by Yu et al. [131] revealed that curcumin decreased CRC cell proliferation by upregulation of lncRNA-NBR2 via the AMPK/mTOR axis. In a concentration-dependent manner, curcumin decreased the viability of A549 cells by enhanced cell apoptosis and diminished cyclin D1 expression. Further studies have shown that the curcumin property occurs through the downregulation of lncRNA-UCA1 and reducing the Wnt and mTOR signaling pathways [134]. In addition, curcumin induces cell death by adjusting the expression of lncRNAs. Dysregulation of lncRNAs can also modify the anticancer effect of curcumin. For example, lncRNA-PANDAR has the same expression in tumor and healthy colorectal cancer tissue. When cells are treated with low doses of curcumin (5µM), PANDAR is overexpressed and prevents the induction of apoptosis by curcumin. Nevertheless, when PANDAR is knocked down by siRNA, cellular senescence decreased and curcumin could induce apoptosis [186].

### 6.7. 3,3′-Diindolylmethane (DIM)

3,3′-Diindolylmethane (DIM), a phytocompound present in various cruciferous vegetables, such as cabbage, broccoli, and kale, is known to possess anticancer activity [187]. The prostate cancer gene expression marker (PCGEM1) is one of the first carcinogenic lncRNAs to be identified and upregulated in many cancers, including cervical carcinoma [188]. PCGEM1 is also upregulated in prostate cancer cells, and its inhibition induces apoptosis and enhances the sensitivity of LNCaP cells to doxorubicin (DOX) [189]. In vitro and in vivo studies by Ho et al. [135] showed that DIM could play an essential role in inhibiting prostate cancer cell growth and decrease their castration resistance by downregulation of PCGEM1 through p54/nrb, which is a transcriptional regulator of PCGEM1. DIM has an inhibitory effect on the p54/nrb function and prevents its interaction with the PCGEM1 promoter. In another study, Zinovieva et al. [136] investigated the effect of DIM on colon cancer cells. They found that HOTAIR and CCAT1-L were increased in the tissues and cell lines of this cancer, and treatment of HT-29 and HCT-116 cells with DIM decreased these two lncRNAs.

### 6.8. Epigallocatechin-3-Gallate (EGCG)

EGCG is another natural polyphenol compound that has cancer-preventive and therapeutic properties [190,191]. In vitro and in vivo studies reveal that EGCG could induce apoptosis and reduce proliferation by activating caspases protease, suppressing NF-κB activation, and adjusting the expression of cell cycle regulatory proteins [192]. Cisplatin is one of the most useful drugs in the treatment of many cancers, including NSCLC. Copper transporter protein 1 (CTR1) is a membrane transport protein that plays an essential role in cisplatin sensitivity [193]. Unfortunately, it has been found that drug resistance leads to treatment failure in some patients. Previous studies have shown that EGCG could increase CTR1 expression through various mechanisms. In vitro and in vivo studies by Chen et al. [25] showed that ROS produced by EGCG could upregulate lncRNA-NEAT1 expression and consequently, decreased p-ERK1/2. EGCG, by changes in NEAT1/p-ERK, could increase CTR1 expression, improve cisplatin sensitivity, and decrease cell proliferation [25]. Further studies showed that NEAT1 functions like a sponge for miR-98-5p and could downregulate the expression of miR-98-5p [194]. CTR1 is one of the target genes of miR-98-5p, and considering miR-98-5p has specific complementary binding sites for NEAT1, an upregulation in NEAT1 increases CTR1 expression. Therefore, in vivo and in vitro use of EGCG induces cisplatin sensitivity via the NEAT1/miR-98-5p/CTR1 axis [137]. 

EGCG can also have a synergistic effect on doxorubicin by modulating SOX2OT V7 and induce osteosarcoma cell death. SOX2OT V7 is a carcinogenic lncRNA that is upregulated in osteosarcoma tumors compared to adjacent tissues. EGCG exerts its anticancer function by reducing SOX2OT V7 and blocking the Notch3/DLL3 signaling pathway, increasing drug sensitivity and cell death [138]. Previous studies confirmed LINC00511 was overexpressed in various cancers, especially gastric cancer [195]. This lncRNA is also overexpressed in the cell lines and tumor tissue of gastric cancer. The inhibition of LINC00511 reduces invasion, migration, and cell proliferation by upregulation miR-29/b. Hence, EGCG could induce the miR-29b/kdm2 axis by downregulation LINC00511 and reduce gemcitabine-resistant gastric cancer [139]. As LINC00511 has a complementary binding site on miR-29/b, it can directly interact with miR-29c and suppress its expression. Therefore, knockdown of LINC00511 could increase the cytotoxicity of paclitaxel in breast cancer cells [196].

### 6.9. Gambogic Acid 

Gambogic acid (GA), a natural plant compound derived from *Garcinia hanburyi*’s resin and has anticancer, antiviral, antioxidant, and anti-inflammatory properties [197]. GA, via upregulation of the tumor suppressor miR-101 and downregulation of EZH2, could induce apoptosis in bladder cancer cells. Since the EZH2, a catalytic component of PRC2, functions as a histone methyltransferase, EZH2 by epigenetic modification of histone H3 can regulate various genes and has a significant role in cancer progression and development [198]. For the first time, Wang et al. [141] revealed that the lncRNA growth arrest-specific 5 (GAS5) was downregulated in bladder cancer (BC) cells and human tissues. GAS5 plays an essential role in cycle arrest, apoptosis, and regulation of cancer cell survival and is located on chromosome 1q25.1 [199]. Moreover, GA is the positive regulator of GAS5 by upregulation of miR-101, and downregulation of EZH2 could promote apoptosis and decrease BC cells’ viability [141]. Khaitan et al. [200] first identified lncRNA SPRY4 transcript 1 (SPRY4-IT1) as an oncogenic lncRNA in melanoma. Through the downregulation of SPRY4-IT1, GA could promote apoptosis and reduce the proliferation of BC cells [200]. SPRY4-IT1 works as sponges for miR-101-3p, and by downregulation, miR-101 could influence EZH2 expression [140].

### 6.10. Genistein 

Genistein (5,7-trihydroxy isoflavone) is abundant in soybeans and has anti-inflammatory, antioxidant, and antiangiogenic properties [201,202]. Genistein plays an essential role in controlling cancer by inducing apoptosis or suppressing cell cycle, angiogenesis, and metastasis [201]. Treatment of MCF-7 cells with genistein and calycosin induces apoptosis and restrains cell proliferation by reducing p-Akt and HOTAIR expression. However, calycosin was more potent in inhibiting breast cancer cell lines’ growth than genistein [123]. HOTAIR plays an essential role in inducing malignancy, invasion, and proliferation in various cancers, and in contrast, miR-141 in a sequence-specific manner can suppress malignancy in human cancer cells by inhibiting HOTAIR [142]. Genistein could perform its anticancer effect by manipulating the expression of microRNAs. For example, in human renal carcinoma, colorectal adenocarcinoma, and prostate cancer cells, genistein represses HOTAIR by upregulation miR-141 [142]. Lynch et al. [203] showed that genistein via demethylation of the CpG island closest to the promoter of miR-141 could increase miR-141 expression. As miR-141 has a complementary binding site on the sequence of HOTAIR in a sequence-specific manner, it can cleave HOTAIR through the Ago2 complex and suppresses its functions [142]. Chiyomaru et al. [143] conducted in vivo and in vitro studies and revealed that genistein could reduce HOTAIR and upregulated miR-34a in prostate cells to inhibit proliferation, migration, and invasion [143]. miR-34a plays an essential role in increasing apoptosis by targeting many oncogenes. Nevertheless, it has been shown that HOTAIR can act as a sponge for miR-34a-5p, impairing its ability to promotes apoptosis [204]. Colorectal cells are also sensitive to genistein, and the treatment of SW480 cells with 50 µM genistein suppresses viability and migration by downregulation lncRNA-TTTY18, Akt, SGK1, and p38 MAPK [144].

### 6.11. Ginsenosides 

According to previous studies, Ginsenoside is a natural product of steroid glycosides with diverse chemical structures that have several pharmacological effects on the cardiovascular, nervous, immune systems, and cancer [205]. These compounds alter the expression of lncRNAs and other cancer-related genes by epigenetic modifications [206]. lncRNA-C3orf67-AS1 is overexpressed in breast cancer cells, and its suppression via specific siRNA in MCF-7 cells leads to restricted colonization and proliferation by inducing apoptosis. Ginsenoside (Rh2) plays an essential role in breast cancer treatment because it could downregulate C3orf67-AS1 by persuading the hypermethylation of the C3orf67-AS1 promoter in MCF-7 cells [145]. Additionally, treatment of MCF-7 cells with ginsenoside (Rg3) showed that Rg3 was able to downregulate and upregulate the expression of two oncogenic and antioncogenic lncRNAs (RFX3-AS1 and STXBP5-AS1) via hypermethylation and hypomethylation of the promoter, respectively [146]. lncRNA-CASC2 could modulate various signaling pathways, acting as a tumor-suppressing lncRNA. CASC2 is downregulated in many human carcinomas, including pancreatic cancer [207]. In vitro and in vivo studies in gemcitabine-resistant pancreatic cancer cells have shown that Rg3 can activate PTEN signaling by upregulation of CASC2, thereby reducing cell proliferation and tumor growth [147]. Rg3 (IC_50_: 50 µM, 24 h) is also able to reduce the expression of Bcl-2, vimentin, and CCND1, and increase the expression of Bax, p53, cleaved-caspase-3 genes in CRCs by downregulation of CCAT1, thereby reducing cell proliferation, invasion, and migration [148]. Additionally, Rg3 can inhibit the Warburg effect in ovarian cancer cells using the H19/miR-324-5p/PKM2 axis. Indeed, Rg3 indirectly reduced PKM2 and the Warburg effect by restraining lncRNA-H19 and inhibited ovarian cancer tumorigenesis in vivo and in vitro. In practice, Rg3 removes the inhibitory effect of H19 on miR-324 by reducing H19 expression, allowing miR-324 to exert its inhibitory effect on PKM2 and the Warburg effect [149].

### 6.12. Hyperoside

Hyperoside is a flavonol glycoside compound that has been found to suppress the growth of various cancers using several different mechanisms. It is able to induce apoptosis and inhibit proliferation in T790M-positive NSCLC cell lines; upregulate the expression of forkhead box protein O1 (FoxO1); downregulate the long lncRNA colon cancer-associated transcript 1 (CCAT1) levels in T790M-positive NSCLC cell lines; and repress the tumor growth of T790M-positive NSCLC xenografts [150].

### 6.13. Luteolin

Luteolin is a natural antioxidant that has antitumorigenic effects and is therefore able to significantly inhibit thyroid cancer growth [208]. By reducing the expression of BRAF-activated long noncoding RNA (BANCR), it further results in downregulation of downstream oncogenic signaling and TSHR. Furthermore, BANCR/TSHR signaling overexpression can significantly inhibit the luteolin antitumor impacts on thyroid cancer in vivo and in vitro and, as a result, can act as an essential anticancer agent for thyroid cancer through suppressing the BANCR/TSHR pathway [151].

### 6.14. Polydatin

Polydatin targets particular molecular modifications in several cancers and demonstrates promising results in treatment of osteosarcoma [209]. Research shows it might significantly induce apoptosis and suppress proliferation in doxorubicin-resistant osteosarcoma cell lines via TUG1 mediated inhibition of Akt pathway in a concentration-dependent manner. As a result, inhibition of TUG1/Akt pathway is crucial for polydatin in treatment of doxorubicin-resistant osteosarcoma cell lines. Polydatin has also been shown to significantly suppress tumor growth in vivo by reducing TUG1 expression and Akt phosphorylation level [152,210]. An in vivo study further established the anticancer impact of polydatin and its related mechanisms.

### 6.15. Quercetin

Quercetin is a flavonoid compound which has been indicated to suppress cancer in vivo and in vitro virtually [211]. MALAT1 can induce the process of EMT, invasion, proliferation, migration, and inhibition of cell apoptosis by activating the PI3K/Akt pathway. Therefore, quercetin can prevent the growth and development of cancer by inhibiting MALAT1. Moreover, quercetin treatment can significantly decrease tumor volume, weight, and MALAT1 expression in vivo. However, overexpression of MALAT1 reduced the impact of quercetin in xenograft models [153]. It was speculated that quercetin treatment suppressed the growth of PC cells via downregulating MALAT1.

### 6.16. Resveratrol

Resveratrol belongs to a group of natural polyphenol compounds called stilbenes [212], which regulate the expression of lncRNAs-PCAT29, NEAT1, MALAT1, and AK001796 and plays an essential role in cancer inhibition [154,155,156,158]. The anticancer and chemopreventive effects of resveratrol are due to its interaction with various molecular pathways and multiple target genes. Resveratrol upregulates PCAT29 and PDCD4 by inhibiting the IL-6/STAT3/miR-21 axis in prostate cancer, giving the potential to decrease cell proliferation and tumorigenesis. PCAT29 is an essential lncRNA in suppressing prostate cancer and downregulated in tumor tissue and cell lines of this cancer [154]. Wang et al. [213] showed that resveratrol could reduce miR-21 expression by downregulation of p-Smad2/3, TGF-β1, c-Fos, c-Jun, p-c-Jun p-p38, p-JNK, p-ERK and suppressive effects on MAPK/AP-1 pathway. NEAT1 is one of several lncRNAs that is overexpressed in multiple myeloma. Resveratrol can reduce invasion, proliferation, and migration of melanoma cells with downregulation of NEAT1. Additionally, resveratrol inhibits survivin, β-catenin, MMP-7, c-Myc, and the Wnt/β-catenin signaling pathway [155]. Resveratrol exerts its antitumor effect on colorectal cancer cells by several mechanisms. One is the downregulation of c-Myc and MMP-7 genes by reducing β-catenin in the nucleus. It can also reduce MALAT expression with a negative effect on its promoter. Therefore, in a concentration-dependent manner, by downregulation, MALAT1 reduces the Wnt/β-catenin pathway’s activity [156]. lncRNA-AK001796 is overexpressed in tumor tissues and lung cancer cells. Inhibition of AK001796 with siRNA leads to decreased proliferation, colonization, and tumor growth [158]. Yang et al. [158] revealed that the chemopreventive agent resveratrol could downregulate AK001796 in A549 cells and induce cell cycle arrest (G0/G1) and cell apoptosis.

### 6.17. Sanguinarine

Sanguinarine, a benzo phenanthridine alkaloid, showed anticancer impacts through suppressing invasion, cell viability, and migration and increasing apoptosis [214]. Furthermore, sanguinarine induced the expression of CASC2 in SKOV3 epithelial ovarian tumor cell lines. lncRNA-CASC2 and EIF4A3 significantly reduced and increased in ovarian cancer cells and tissues, respectively. EIF4A3 knockdown reversed the impacts of sanguinarine plus silencing of CASC2. Moreover, sanguinarine significantly suppressed the activation of the NF-κB or PI3K/AKT/mTOR pathway, which was reversed through silencing of CASC2 [159].

### 6.18. Silibinin

Silibinin is the main active constituent of silymarin, which has been indicated to have significant anticancer impacts in various malignancies. Silibinin can significantly induce apoptosis and inhibit invasion, migration, proliferation by downregulating the PI3K/Akt and STATs in various cancers [215]. Silibinin downregulated the PI3K/Akt and actin cytoskeleton pathways in T24 and UM-UC-3 human bladder tumor and cell lines. The silibinin inhibited trimethylated histone H3 lysine 4 and acetylated H3 at the promoter of KRAS. Moreover, silibinin targets lncRNAs, such as ZFAS1 and HOTAIR, which are identified as oncogenic factors in several cancers [160].

### 6.19. Sulforaphane

Sulforaphane, obtained from various vegetables, such as broccoli, brussels sprouts, cauliflower, kale, cabbage, watercress, and bok choy, can suppress invasion, migration, cell growth and induce apoptosis in tumor cells [216]. It also reduces the expression of tumor-associated lncRNAs. Sulforaphane significantly altered the expression of around 100 lncRNAs and normalized the differential expression of various lncRNAs in tumor cells. SFN-mediated modifications in the expression of lncRNA associated with genes regulate the metabolism, signal transduction, and cell cycle [161]. 

## 7. Mechanism of Actions of Phytochemicals in the Regulation of lncRNAs in Cancer

Epigenetic mechanisms, such as genetic changes, play a significant role in the onset and development of cancer. For example, promoter hypermethylation or hypomethylation causes many cancer-related genes to be turned on or off, respectively [217]. Phytochemicals can play an essential role in the treatment of cancers. These compounds act as epigenetic modifiers (DNA methylation and histone modifications) and could regulate the expression of coding genes and non-coding RNAs, such as miRNAs and lncRNAs [218]. Phytochemicals, such as resveratrol, DIM, polydatin, luteolin, and curcumin exert their therapeutic and prophylactic activity against cancer by inhibiting the expression of specific genes such as histone deacetylases (HDACs) and DNA methyltransferases (DNMTs) [218]. By doing this, phytochemicals can alter the expression of miRNAs, lncRNAs, and transcription factors [219,220,221,222]. EZH2 and SUZ12 are the primary epigenetic mediators expressed at high levels in cancer stem cells. Curcumin reduced the expression of PVT1 lncRNA by inhibiting EZH2 and SUZ12 in pancreatic ductal adenocarcinoma cells [125]. P54/nrb, a ubiquitously expressed protein, performs numerous roles, including RNA splicing, transcriptional regulation, and nuclear retention [223]. DIM downregulated lncRNA-PCGEM1 by repressing binding of p54/nrb to the PCGEM1 promoter [135]. Sulforaphane increased the expression of Loc344887 in HCT116 colon cancer cells by acting on the nuclear factor erythroid 2-related factor 2 (*NFE2L2* or *Nrf2*) [162]. Another anticancer mechanism of phytochemicals involves regulation of cellular signal transduction pathways, such as Akt, Wnt/β-catenin, MAPK, and JAK/STAT, that could alter gene expression in cancer cells [224]. Curcumin inhibited H19 expression via downregulation of c-Myc [24]. Various phytochemicals can target and regulate the expression of lncRNAs via their action on miRNAs. For example, curcumin by upregulation of miR-185 and miR-29a can decrease the expression of DNMT1, DNMT 3A, and DNMT 3B to upregulate MEG3 in hepatocellular cancer cells [132]. Moreover, curcumin via upregulation of miR-145 or berberine and genistein via upregulation of miR-34a-5p could decrease the lncRNA-ROR and HOTAIR, respectively [27,127,143]. Genistein can upregulate miR-141 expression, and miR-141 has a binding site in HOTAIR. Therefore, miR-141, via a sequence-specific manner, can suppress HOTAIR expression in renal carcinoma and colorectal adenocarcinoma cells [142] (Figure 3).

## 8. Phytochemical-Based Regulation of lncRNAs in Cancer Precision Medicine

Precision medicine plays an essential role in determining prognosis and response to treatment of various diseases, including cancer [225]. The inhibitory effect of gefitinib and erlotinib on epidermal growth factor receptors has led to their use in treating NSCLC patients. However, NSCLC cancer cells with D761Y and T790M point mutations in their EGFR sequence are resistant to these two drugs [226]. Therefore, physicians should first use one of the molecular methods and analyze the EGFR sequence to avoid prescribing gefitinib and erlotinib for patients carrying these point mutations. Moreover, the high expression level of lncRNA EGFR-AS1 in head and neck squamous-cell cancers increases their resistance to gefitinib and erlotinib [227]. Therefore, gefitinib and erlotinib therapy will not be useful in patients with overexpression of EGFR-AS1. By regulating the expression of lncRNAs, bioactive phytocompounds have the potential to treat and manage various malignant diseases. For example, baicalin, berberine, curcumin, and EGCG induce cell cycle arrest and apoptosis and decrease the proliferation of cancer cells by overexpression of NKLA, PAX8-AS1, CASC-2, NBR2, XIST, NEAT-1, SOX2OT-V7, respectively [25,26,118,119,129,131,138]. However, it should be noted that this anticancer property of natural compounds occurs when, first, cells can express these lncRNAs, and second, the sequence of these lncRNAs does not have any loss-of-function mutation; otherwise, they will not be useful. EGCG through upregulation of LINC00511 and upregulation miR-29 could reduce migration, invasion, and cell proliferation of gastric cancer cells. However, if miR-29b has a loss-of-function mutation in its sequence, its overexpression by EGCG will not inhibit the invasion, proliferation, and migration of cancer cells [139]. 

## 9. Conclusions, Current Challenges, and Future Perspectives

lncRNAs perform an essential role in regulating cellular processes, and their dysregulation can lead to uncontrolled growth or cell death. This has led to increases in oncogenic lncRNAs and decreases in the expression of tumor suppressor lncRNAs. Phytochemicals are one of the gifts of nature that help to maintain good health and cure disease. The results of several experimental studies have shown that phytochemicals act as a safe and effective anticancer agent. Among all mechanisms associated with phytochemicals, the lncRNA-regulatory effects are prominent and have been demonstrated by previous studies. In this regard, baicalin, berberine, curcumin, EGCG, genistein, ginsenosides, and resveratrol have been identified as the leading natural compounds with both anticancer and lncRNA-regulatory properties (Figure 4).

Although several experimental studies have indicated the benefits of phytochemicals in cancer treatment, their clinical evaluation has been faced with several principal challenges. Various reports suggest that effective doses obtained from in vivo studies may adversely affect future clinical trials. It seems that more experimental studies, especially in laboratory animal models, should be conducted to guide future clinical studies. Together, the low bioavailability of some phytochemicals has a disruptive impact on clinical outcomes. While nanoformulation of phytochemicals has emerged as an alternative strategy to improve the bioavailability of phytochemicals, nanomaterials’ application is still limited for clinical goals [228]. 

Moreover, most of these phytochemicals should be consumed in high doses to be effective against cancer, and achieving these doses is not possible only through diet. Cancer is a complex disease with different etiologies, and its occurrence is due to the dysregulation of several signaling pathways. A useful idea to overcome this deficiency is to combine standard anticancer drugs with phytochemicals or combine several phytochemicals to and use them simultaneously [229]. Previous studies have clearly shown that the synergistic effects of piperine, sulforaphane, and thymoquinone on other standard anticancer drugs such as doxorubicin, paclitaxel, 5-fluorouracil, and gemcitabine make them more useful and cytotoxic in controlling cancer cells [230].

Phytochemicals combined with the synergistic effect could inhibit the invasion, metastasis, and proliferation of malignant cells more than a phytochemical, these making them practical options for future cancer treatment [231]. A combination of phytochemicals, such as piperine with curcumin, piperine with thymoquinone, and sulforaphane with genistein, could lower the IC_50_ values for individual agents, and also causes inhibition of proliferation or induction of apoptosis more effectively [230]. Previous studies have shown that various phytochemicals, such as resveratrol, sulforaphane, quercetin, and genistein, can inhibit prostate cancer cells by upregulation PCAT29 and downregulation LINC01116, MALAT1, and HOTAIR, respectively. As they can affect various molecular pathways, their combination cold improves their effectiveness in treating this cancer [143,153,154,161]. 

Despite the potential of natural compounds to inhibit cancer, their low solubility in water, poor bioavailability, and the need for high doses have limited their clinical efficacy. The combination of nanoscience and chemistry has shown that nano-based formulations could reduce the dosage and increase cancer treatment effectiveness by increasing the bioavailability, solubility, and specific targeting of phytochemicals [232]. In addition, bioactive phytochemicals are less expensive compared to synthetic agents and are widely available; these compounds have limited side effects and are beneficial to human health. Therefore, natural compounds can be a novel and alternative treatment strategy for cancer treatment and can improve tumor cells’ sensitivity to standard adjuvant therapies such as chemotherapy and radiotherapy.

## Figures and Tables

**Figure 1 cancers-13-01274-f001:**
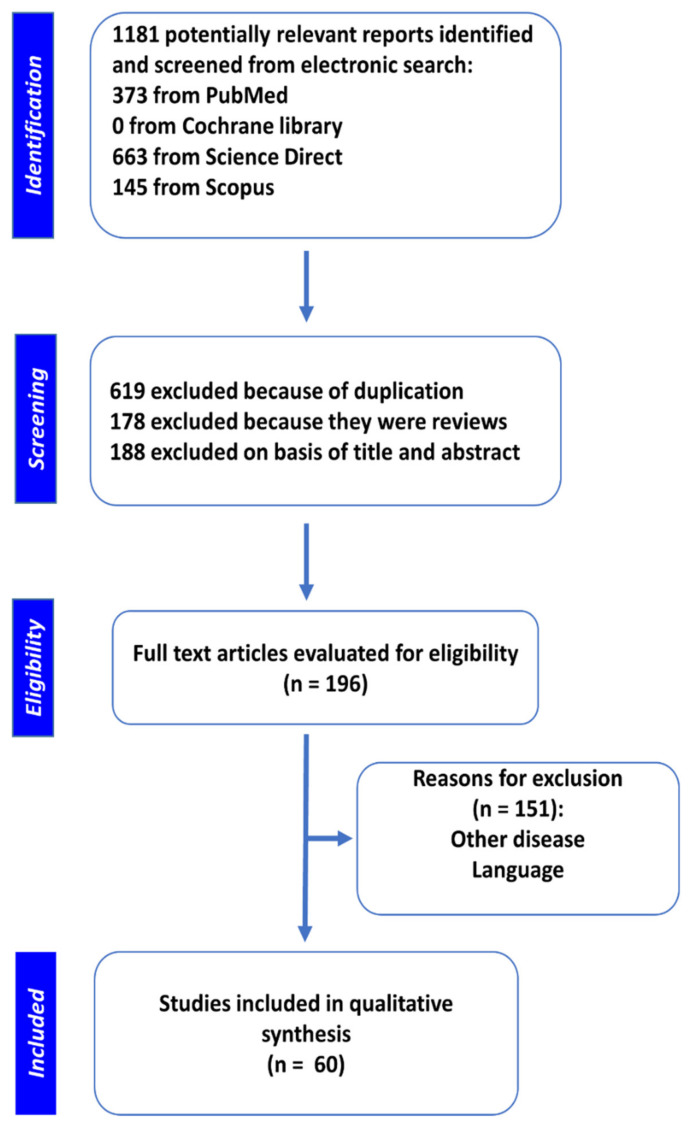
Flow diagram of literature exploration and collection.

**Figure 2 cancers-13-01274-f002:**
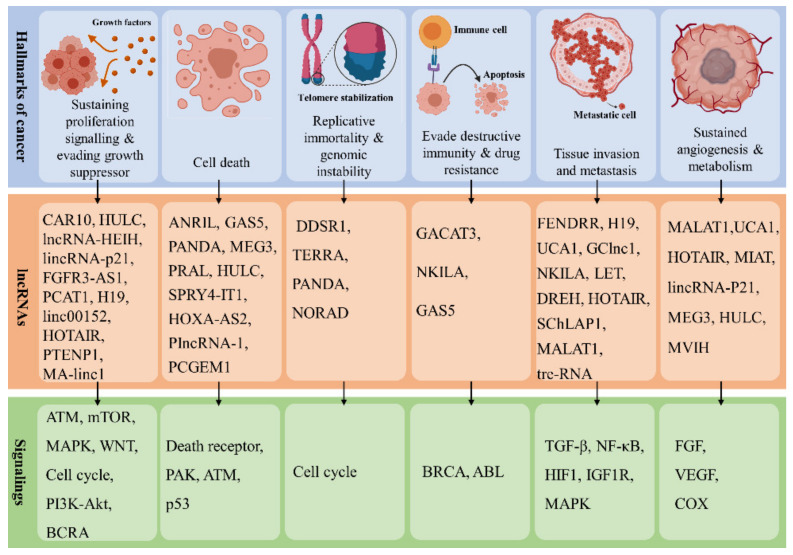
A schematic view of controlling cancer hallmarks and associated signaling pathway by lncRNAs.

**Figure 3 cancers-13-01274-f003:**
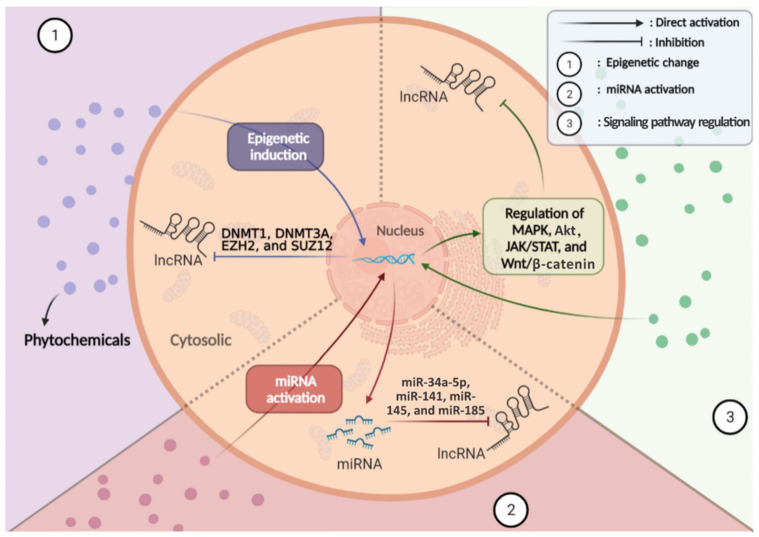
Molecular mechanisms action of phytochemicals for regulation of lncRNAs in cancer (Created using resources at Biorender.com).

**Figure 4 cancers-13-01274-f004:**
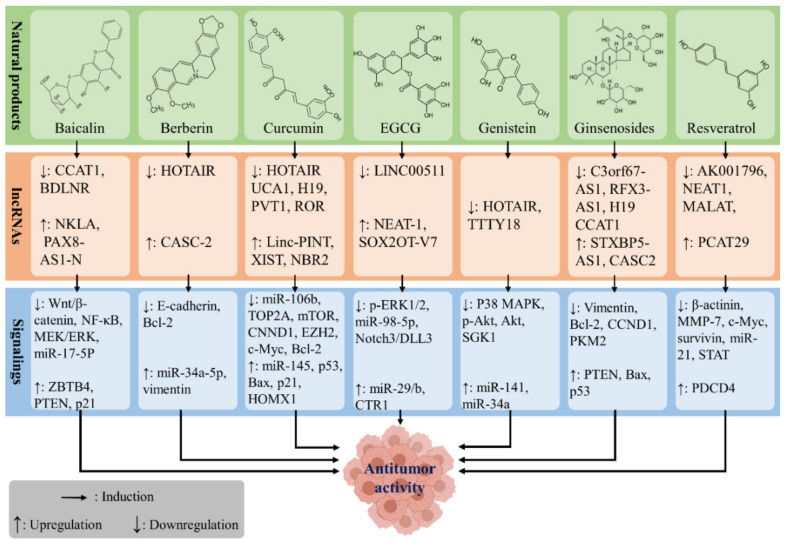
The potential anticancer effect of phytochemicals via regulation of the expression of lncRNAs.

**Table 1 cancers-13-01274-t001:** Summary of various well-known cancer-related lncRNAs, their molecular functions, and cellular outcomes.

lncRNAs	Type	Related Cancer	Molecular Function	Outcome	Reference
ANRIL	Oncogenic	Bladder, breast, colon, gastric, head and neck, leukemia, liver, lung, osteosarcoma, ovarian, pancreas, prostate, renal, SNC, and uterine	Positive correlation MMP3, caspse-9, caspase-3, Bcl-2, E2F1, c-Myc, and miR-122-5p; Negative correlation p15, p16, TIMP2, Bax, miR-99a, and miR-449a	Induction of cell proliferation, cell cycle, migration, metastasis, and EMT; Inhibition of apoptosis and autophagy cell death	[52,53,54,55]
CCAT1/2	Oncogenic	Bladder, breast, colon, gastric, head and neck, leukemia, liver, lung, osteosarcoma, ovarian, pancreas, prostate, renal, SNC, and uterine	Positive correlation GAC	Induction of reprogramming of energy metabolism	[56,57]
CRNDE	Oncogenic	Bladder, breast, colon, gastric, leukemia, liver, lung, ovarian, pancreas, renal, SNC, and uterine	Positive correlation GLUT4; Negative correlation Insulin, IGF-I, and IGF-II	Induction of cell proliferation, migration, metastasis, and reprogramming of energy metabolism	[58,59,60]
GAS5	Tumor suppressor	Bladder, breast, colon, gastric, leukemia, liver, lung, osteosarcoma, ovarian, pancreas, prostate, renal, SNC, and uterine	Positive correlation PTEN and p21; Negative correlation ERK1/2, NF-κB, CDK6, E2F1, cyclinD1, Vimentin, and MMP2	Induction of autophagy cell death; Inhibition of cell survival, proliferation, migration, metastasis, and cell cycle	[12,13,17,61]
H19	Oncogenic	Bladder, breast, colon, gastric, head and neck, leukemia, liver, lung, osteosarcoma, ovarian, pancreas, prostate, renal, SNC, and uterine	Positive correlation c-Myc, cyclinA2, CDK4, cyclinB1, cyclin D1, and cyclin E1; Negative correlation RB, EGFR, p21, and IGF-II	Induction of cell proliferation, cell cycle, migration, metastasis, EMT, tumor angiogenesis, and immune escape; Inhibition of apoptosis and autophagic cell death	[17,62,63,64]
HOTAIR	Oncogenic	Bladder, breast, colon, gastric, head and neck, leukemia, liver, lung, osteosarcoma, ovarian, pancreas, prostate, renal, SNC, and uterine	Positive correlation cyclin D1, cyclin E1, CDK4, CDK2, E2F1, P38, Bcl-2, NOTCH1, β-catenin, N-cadherin, Vimentin, Snail, Twist, MMP-9, MMP-2, MMP-3, FGF1, VEGFA, Ang2, and GLUT1; Negative correlation p53, p21, p16, PIK3R3, caspase-9, and caspase-3	Induction of cell proliferation, cell cycle, invasion, metastasis, EMT, tumor angiogenesis, and immune escape; Inhibition of apoptotic and autophagic cell death	[17,65,66]
HULC	Oncogenic	Bladder, breast, colon, gastric, head and neck, leukemia, liver, lung, osteosarcoma, ovarian, pancreas, prostate, SNC, and uterine	Positive correlation ZEB1, ZO-1, LC3-II/LC3-I, pmTOR, E2F1, and Snail; Negative correlation E-cadherin	Induction of cell proliferation, migration, and metastasis; Inhibition of apoptotic cell death	[17,18]
MALAT1	Oncogenic	Bladder, breast, colon, gastric, head and neck, leukemia, liver, lung, osteosarcoma, ovarian, pancreas, prostate, renal, SNC, and uterine	Positive correlation CDK4, ZEB2, slug, β-catenin, N-cadherin, vimentin, Twist, MMP1, MMP-9, VEGF, TGFA, and TGF-β Negative correlation BAX, E-cadherin, MMP19, TIMP-3, and miR-200s	Induction of cell cycle, cell proliferation, EMT, differentiation, migration, metastasis, chemoresistance and tumor angiogenesis; Inhibition of DNA damage, apoptosis and autophagy cell death	[67,68,69,70,71]
MEG3	Tumor suppressor	Bladder, breast, colon, gastric, head and neck, leukemia, liver, lung, osteosarcoma, ovarian, pancreas, prostate, renal, SNC, and uterine	Positive correlation p53, caspase-3, procaspase-9, cyt. c, and Bax; Negative correlation PI3K, Akt, mTOR cyclin D1, cyclin B1, CDK1, and Bcl-2	Induction of apoptotic and autophagic cell death; Inhibition of cell proliferation, invasion, metastasis, and cell cycle	[17,72,73,74]
PVT1	Oncogenic	Bladder, breast, colon, gastric, head and neck, leukemia, liver, lung, osteosarcoma, ovarian, pancreas, prostate, renal, SNC, and uterine	Positive correlation EZH2, c-Myc, CD115, FGF2, and HIF-1α; Negative correlation miR-31, miR-152, miR-186, and miR-195	Induction of cell proliferation, migration, metastasis, cell cycle, and angiogenesis; Inhibition of apoptotic cell death	[17,75]
TERRA	Tumor suppressor	Bladder, breast, colon, gastric, head and neck, leukemia, liver, lung, osteosarcoma, ovarian, pancreas, prostate, renal, SNC, and uterine	Negative correlation TRF2	Induction of apoptotic cell death; Inhibition of cell proliferation, invasion, metastasis, and cell cycle	[76,77,78]
UCA1	Oncogenic	Bladder, breast, colon, gastric, head and neck, leukemia, liver, lung, osteosarcoma, ovarian, pancreas, prostate, renal, SNC, and uterine	Positive correlation PI3K, AMPK, cyclinD1, ZEB1, ZEB2, N-cadherin, Vimentin, Snail, β-catenin, MMP-7, and FGFR1; Negative correlation p27, E-cadherin, and MMP-14	Induction of cell proliferation, migration, metastasis, cell cycle, EMT, and reprogramming of energy metabolism; Inhibition of cellular apoptosis	[19,79,80]

**Table 2 cancers-13-01274-t002:** Anticancer therapeutic effects of phytochemicals by regulation of lncRNAs and their target genes in various in vitro cancer models.

Phytochemical	Source Plant	Cancer Type	IC_50_, Exposure Time	Target lncRNAs	Target Genes	Biological Functions	Reference
Anacardic acid	*Anacardium occidentale* L. (Anacardiaceae)	Breast cancer (MCF-7 and MDA-MB-231 cells)	13.5 µM and 35.0 µM, 6 h	CFLAR-AS1, UBL7-AS1 and MIR210HG↓	SCD, INSIG1 and TGM2↓PDK4, GPR176 and ZBT20↑	Inhibits proliferation	[116]
Baicalein	*Scutellaria baicalensis* Georgi	Melanoma (A375 and SK-MEL-28 cells)	50 µM, 24 h	CCAT1↓	Wnt-3a, β-catenin, MEK and ERK↓	Inhibits proliferation, invasion, migration and promotes apoptosis	[117]
		Cervical cancer (HeLa and SiHa cells)	100 µM, 24 h	lncRNA- BDLNR↓	PI3K/Akt pathway↓	Inhibits tumor growth, migration and proliferation	[23]
		Hepatocellular carcinoma (SMMC-7721, HCCLM3, Hep3B and HepG2 cells)	12.5 µM, 48 h	NKILA↑	NF-κB signaling↓	Inhibits migration, proliferation and promotes apoptosis	[118]
		Breast cancer (MDA-MB-231 and MCF-7 cells)	50 µM, 48 h	PAX8-AS1-N↑	miR-17-5p↓ ZBTB4, PTEN and CDKN1A↑	Inhibits proliferation and cell death	[119]
Berberis	*Coptis chinensis, Berberis poiretii* Schneid, *Berberis vernae* Schneid, *Berberis wilsoniae* Hemsl	Colorectal cancer (HT29 and HCT116 cells)	40 µM, 48 h	CASC2↑	Bcl-2↓ caspase-3 and caspase-9↑	Inhibits proliferation, viability and promotes apoptosis	[26,120]
		Non-small cell lung carcinoma (A549 and H1975 cells)	25 µM, 24 h	HOTAIR↓	Vimentin↓ E-cadherin and miR-34a-5p↑	Inhibits proliferation, invasion and metastasis	[27]
Bharangin	*Pygmacopremna herbacea* (Roxb), *Premna herbacea*	Breast cancer (MCF-7, MDAMB-231, MDA-MB-453, MDA-MB-468 and T-47D cells)	5 µM, 6 h	H19↓ MEG-3, GAS-5, MHRT and NEAT1↑	NF-κB and Bcl-2↓ Bax↑	Inhibits proliferation and migration and promotes apoptosis and cell cycle arrest	[121]
Calycosin	*Radix astragali*	Breast cancer (MCF-7, T47D, MDA-MB-468, and SKBR3 cells)	16 µM, 24 h	WDR7-7↑	p-SRC, p-EGFR, p-ERK1/2, p-Akt and GPR30↓	Inhibits proliferation and tumor growth	[122]
		Breast cancer (MCF-7 cells)	80 µM, 48 h	HOTAIR↓	p-Akt↓	Inhibits proliferation and promotes apoptosis	[123]
		Nasopharyngeal carcinoma (CNE1, CNE2 and C666-1 cells)	50 µM, 48 h	EWSAT1↓	p- IκBα, p-c-Jun, TRAF6 and p-TAK1↓	Inhibits proliferation and tumor growth	[124]
Curcumin	*Curcuma longa* L.	Pancreatic ductal adenocarcinoma (BxPC3-GemR and Panc1 cells)	8 µM and 20 µM, 48 h	lncRNA-PVT1↓	EZH2 and SUZ12↓	Promotes the sensitivity of BxPC3-GemR cells to gemcitabine	[125]
		Acute lymphoblastic leukemia (LBH-589 cell)	30 µM, 48 h	linc-PINT↑	HOMX1↑	Promotes apoptosis and cell cycle arrest	[126]
		Gastric cancer (GES-1 cells)	50 µM, 48 h	H19↓	Myc, Bcl-2↓ p53 and Bax↑	Promotes apoptosis	[24]
		Prostate cancer stem cells (Du145 and 22RV cells)	46.5 µM, 48 h	lncRNA-ROR↓	Oct4, CDK4 and cyclin D1↓ miR-145↑	Inhibits invasion, proliferation, xenograft growth and cell cycle (G2/M)	[127]
		Renal cell carcinoma	20 µM, 24 h	HOTAIR↓ XIST↑	miR-106b↓ p21↑	Inhibits migration, invasion and proliferation	[128,129]
		Breast cancer (MCF7, MDA-MB231 and SKBR3 cells)	13.5 µM, 48 h	Tusc7 and GAS5↑	-	Inhibits proliferation and promotes apoptosis and cell cycle	[130]
		Colorectal carcinoma (HCT116 and SW480 cells)	10 µM, 24 h	lncRNA-NBR2↑	AMPK/mTOR signaling axis↓	Inhibits proliferation	[131]
		Hepatocellular cancer (HepG2 and HuH-7 cells)	23 µM, 48 h	MEG3↑	DNMT1, DNMT3A DNMT3B↓ miR-29a and miR-185↑	Inhibits proliferation	[132]
		Ovarian cancer (OVCAR-3 and SKOV3 cells)	1 µM, 36 h	MEG3↑	miR-214↓	Inhibits cisplatin resistance	[133]
		Lung cancer (A549 cells)	1 µM, 24 h	lncRNA-UCA1↓	Cyclin D1 and Wnt/mTOR signaling↓ caspase-3 and caspase-9↑	Inhibits proliferation and promotes apoptosis	[134]
DIM	*Brassica* vegetables, Brussels sprouts, Cruciferous vegetables	Prostate cancer (LNCaP and CWR22Rv1 cells)	20 µM, 48 h	PCGEM1↓	p54/nrb↓	Inhibits castration resistance	[135]
		Colon cancer (HT-29 and HCT-116 cells)	30 µM, 72 h	HOTAIR, CCAT1-L↓	-	-	[136]
EGCG	*Camellia sinensis* (L.) Kuntze (Theaceae)	Non-small cell lung carcinoma (A549 cells and H460 cells)	20 µM, 24 h	lncRNA-NEAT↑	p-ERK1/2 and miR-98-5p↓	Promotes cisplatin sensitivity	[137]
		Osteosarcoma (U2OS and SaoS2 cells)	20 μg/mL, 24 h	SOX2OT V7↓	Notch3/DLL3 signaling↓	Inhibits stemness and autophagy and promotes death	[138]
		Gastric cancer (AGS and SGC7901 cells)	100 μM, 48 h	LINC00511↓	miR-29b↑	Inhibits invasion, proliferation, migration and gemcitabine resistance	[139]
Gambogic acid	*Garcinia hurburyi*	Bladder cancer (EJ, UMUC3 and T24T cells)	1 μM, 48 h	SPRY4-IT1↓	EZH2↓ miR-101↑	Inhibits proliferation, migration and invasion, and promotes apoptosis	[140]
		Bladder cancer (T24 and EJ cells)	2 μM, 48 h	GAS5↑	EZH2↓	Inhibits viability, xenograft growth and promotes apoptosis	[141]
Genistein	*Genista tinctoria* L., *Glycine max* L.	Breast cancer (MCF-7 cells)	80 μM, 48 h	HOTAIR↓	p-Akt↓	Inhibits proliferation and promotes apoptosis	[123]
		Renal carcinoma, colorectal adenocarcinoma and prostate cancer	25 μM, 96 h	HOTAIR↓	miR-141↑	Inhibits proliferation and promotes apoptosis	[142]
		Prostate cancer (PC3 and DU145 PCa cells)	25 μM, 96 h	HOTAIR↓	miR-34a↑	Inhibits proliferation, migration and invasion	[143]
		Colorectal cancer (SW480 cells)	50 μM, 48 h	lncRNA- TTTY18↓	Akt, SGK1 and p38 MAPK↓	Inhibits proliferation and migration	[144]
Ginsenosides	*Panax ginseng, P. quinquefoliu, P. vietnamensis, P. japonicas, P. notoginseng*	Breast cancer (MCF-7 cells)	30 μM, 24 h	C3orf67-AS1↓	-	Inhibits colonization and proliferation and promotes apoptosis	[145]
		Breast cancer (MCF-7 cells)	20 μM, 24 h	RFX3-AS1↓ STXBP5-AS1↑	RFX3, SLC1A1, PUM3 and STXbp5↓ GRM1↑	Inhibits colonization	[146]
		Pancreatic cancer (Panc-1 and SW1990 cells)	50 μM, 24 h	CASC2↑	PTEN signaling↑	Inhibits proliferation and tumor growth	[147]
		Colorectal cancer (Caco-2 cells)	50 µM, 24 h	CCAT1↓	Bcl-2, vimentin and CCND1↓ Bax, p53 and caspase-3↑	Inhibits proliferation, invasion and migration	[148]
		Breast cancer (MCF-7 cells)	80 µM and 40 µM, 24 h	lncRNA-H19↓	PKM2↓	Inhibits tumorigenesis and the Warburg effect	[149]
Hyperoside	*Artemisia capillaris, Apocynum venetum*	Non-small-cell lung carcinoma (NCI-H1975 cells)	87.4 µM, 48 h	CCAT1↓	FoxO1↑	Inhibits proliferation, xenografts growth and promotes apoptosis	[150]
Luteolin	*Passiflora edulis,* *Taraxacum officinale*	Thyroid carcinoma (IHH-4, FTC-133 and 8505C cells)	10 µM, 24 h	BANCR↓	CCND1, p-CREB, PCNA and TSHR signaling↓	Inhibits cell cycle, proliferation and xenograft growth	[151]
Polydatin	*Polygonum cuspidatum* Sieb. et Zucc. (Polygonaceae) *Fallopia japonica* (Houtt.)	Osteosarcoma (Saos-2 and MG-63 cells)	150 µM, 48 h	TUG1↓	p-Akt↓	Inhibits proliferation, tumor volume and tumor weight and promotes apoptosis	[152]
Quercetin	*Allium cepa* L. (Amaryllidaceae)	Prostate cancer (PC-3 cells)	50 µM, 48 h	MALAT1↓	N-cadherin, p-Akt and Bcl-2↓ E-cadherin and Bax↑	Inhibits proliferation, migration, invasion, xenografts growth and EMT and promotes apoptosis	[153]
Resveratrol	Grapes, blueberries, *Morus alba* L., *Polygonum cuspidatum* Sieb. et Zucc., *Rubus idaeus* L.	Prostate cancer (DU145 and LNCaP cells)	25 µM, 24 h	PCAT29↑	IL-6, STAT3 and miR-21↓ PDCD4↑	Inhibits proliferation and tumorigenesis	[154]
		Multiple myeloma (U266 and LP1 cells)	40 µM, 72 h	NEAT1↓	Survivin, β-catenin, MMP-7 and c-Myc↓	Inhibits migration, proliferation and invasion	[155]
		Colorectal cancer (LoVo and HCT116 cells)	50 µM, 48 h	MALAT1↓	c-Myc, MMP-7 and β-catenin↓	Inhibits migration, proliferation and invasion	[156]
		Breast cancer (LTED and MCF-7 cells)	50 µM, 24 h	u-Eleanor↓	ER gene↓	Inhibits cell growth	[157]
		Lung cancer (A549 cells)	40 µM, 48 h	AK001796↓	-	Promotes cell cycle arrest (G0/G1) and apoptosis	[158]
Sanguinarine	*Sanguinaria canadensis* L. (bloodroot)	Ovarian cancer (SKOV3 cells)	5 µM, 48 h	CASC2↑	NF-κB, PI3K, p-Akt↓	Inhibits viability, migration and invasion and promotes cell apoptosis	[159]
Silibinin	*Silybum marianum* (L.) Gaertn. (Asteraceae)	Bladder cancer (T24 and UM-UC-3 cells)	10 µM, 48 h	HOTAIR and ZFAS1↓	EGFR, SOS1, Ras, PAK1, DDR1, H3K4 and p-Akt↓	Inhibits proliferation, migration and invasion and promotes apoptosis	[160]
Sulforaphane	*Brassica oleracea*	Prostate cancer (PC-3 and LNCaP cells)	15 µM, 24 h	LINC01116↓	MAP1LC3B2 and H2AFY↑	Inhibits proliferation and clonogenic survival	[161]
		Colon cancer (HCT116 and HT29 cells)	15 µM, 24 h	Loc344887↑ (NMRAL2P)	-	Inhibits proliferation colony formation and migration	[162]

↓, downregulation; ↑, upregulation; IC_50_, half maximal inhibitory concentration; DIM, 3,3′-diindolylmethane; EGCG, epigallocatechin-3-gallate.

**Table 3 cancers-13-01274-t003:** In vivo regulatory effects of phytochemicals on lncRNAs in various mouse xenograft tumor models.

Phytochemical	Cancer Type	Dose	Target lncRNAs	Biological Functions	Reference
Baicalein	Cervical cancer	10 mg/kg/day	lncRNA- BDLNR↓	Inhibits tumor growth	[23]
	Hepatocellular carcinoma	10 mg/kg/day	NKILA↑	Inhibits tumor growth	[118]
	Breast cancer	10 mg/kg/day	PAX8-AS1-N↑	Inhibits tumor growth	[119]
Berberis	Non-small cell lung carcinoma	25 mg/kg/day	HOTAIR↓	Inhibits tumor growth	[27]
Calycosin	Breast cancer	55 mg/kg/day	WDR7-7↑	Inhibits tumor growth	[122]
	Nasopharyngeal carcinoma	60 mg/kg/day	EWSAT1↓	Inhibits tumor growth	[124]
Curcumin	Pancreatic ductal adenocarcinoma	100 mg/kg/day	lncRNA-PVT1↓	Inhibits tumor growth	[125]
DIM	Prostate cancer	20 mg/kg/day	PCGEM1↓	Inhibits tumor growth	[135]
EGCG	Non-small cell lung carcinoma	20 mg/kg/day	lncRNA-NEAT1↑	Inhibits tumor growth	[137]
	Osteosarcoma	30 mg/kg/day	SOX2OT V7↓	Inhibits tumor growth	[138]
Genistein	Colorectal cancer	30 mg/kg/day	lncRNA-TTTY18↓	Inhibits tumor growth	[144]
Ginsenosides	Pancreatic cancer	40 mg/kg/day	CASC2↑	Inhibits tumor growth	[147]
Hyperoside	Non-small-cell lung carcinoma	25 mg/kg/day	CCAT1↓	Inhibits tumor growth	[150]
Luteolin	Thyroid carcinoma	50 mg/kg/day	BANCR↓	Inhibits tumor growth	[151]
Polydatin	Osteosarcoma	150 mg/kg/day	TUG1↓	Inhibits tumor growth	[152]
Quercetin	Prostate cancer	75 mg/kg/day	MALAT1↓	Inhibits tumor growth	[153]

↓, downregulation; ↑, upregulation; DIM, 3,3′-diindolylmethane; EGCG, epigallocatechin-3-gallate.

## Data Availability

No new data were created or analyzed in this study. Data sharing is not applicable to this article.

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
