# Peer review of "Regulation of Long Non-Coding RNAs by Plant Secondary Metabolites: A Novel Anticancer Therapeutic Approach"

_cancers, 2021, doi:10.3390/cancers13061274_

Round 1

Reviewer 1 Report

This review is well written and informative. Specific comments:

  • In Table 2, it would be advisable to add the effective concentrations of each phytochemical in those studies cited. This information may facilitate the comparison of the potency of these phytochemicals in regulating lncRNAs, and provide information for assessing their in vivo feasibility as well.
  • It seems all the information in Table 2 was from in vitro studies. Are there any in vivo studies to show the regulatory effects of phytochemicals on lncRNA? Due to the low bioavailability of most phytochemicals, it is a concern whether the in vitro observations could be reproduced in vivo.

Author Response

The authors of this manuscript express their sincere thanks to the reviewer for the critical assessment of this work. The authors have acted upon the recommendations of the reviewers which have resulted in a significant enhancement in the quality of this manuscript. All modifications incorporated in the manuscript are highlighted in red color font. A “point-by-point” response to each and every comment is outlined below.

Comment 1:

This review is well written and informative. Specific comments:

Response:

We thank the reviewer for their expertise, time, and effort for reviewing our manuscript. We are deeply encouraged by the reviewer’s generous comments about the quality of our manuscript. We have addressed the reviewer’s specific comments as indicated below.

Comment 2:

In Table 2, it would be advisable to add the effective concentrations of each phytochemical in those studies cited. This information may facilitate the comparison of the potency of these phytochemicals in regulating lncRNAs, and provide information for assessing their in vivo feasibility as well.

Response:

This is an excellent suggestion. We have redesigned Table 2 by incorporating a new column to provide half maximal inhibitory concentrations (IC50) of various phytochemicals (pages 11-16).

Comment 3:

It seems all the information in Table 2 was from in vitro studies. Are there any in vivo studies to show the regulatory effects of phytochemicals on lncRNA? Due to the low bioavailability of most phytochemicals, it is a concern whether the in vitro observations could be reproduced in vivo.

Response:

The reviewer has made an excellent point. We have introduced a new table (Table 3) to present in vivo studies showing the regulatory effects of phytochemicals on lncRNAs in various mouse xenograft tumor models (page 17). These studies are also discussed under various phytochemicals in section 6. As far as we know, no clinical trial studies have been published in this field to date.

Additionally,

  1. The reference list has been modified as we have added several new references. Special attention is given to conform to the order of references and bibliographic style of the journal.
  2. The entire manuscript has been thoroughly checked and edited to ensure uniform style, organization, and quality.

On behalf of my co-authors, I once again express my sincere thanks to the erudite reviewer for the valuable suggestions and constructive input to improve the quality of our manuscript.

Reviewer 2 Report

Thank you for sending me the research article paper “ Regulation of long non-coding RNAs by plant secondary 2 metabolites: A novel anticancer therapeutic approach” for review in the Cancers. In the article of Kalhori et al., the author discussed the role of LncRNA and phytochemical in cancer prevention and treatment. Topic is interesting and novel. However, there are important points that should be improved.

  1. Author should divide the heading into sub-heading like; 2. Methodology for literature search and selection, 3. lncRNAs: categories and functions and others. It would be helpful for the readers to read and understand.
  2. 4. Role of lncRNA in human cancers: this heading should be divided into several sub-headings. could be divided according to LncRNA or cancer types.
  3. Author should construct a graphical representation of the following heading: Mechanism of action of phytochemicals in the regulation of lncRNAs in cancer.

Author Response

The authors of this manuscript express their sincere thanks to the reviewer for the critical assessment of this work. The authors have acted upon the recommendations of the reviewers which have resulted in a significant enhancement in the quality of this manuscript. All modifications incorporated in the manuscript are highlighted in red color font. A “point-by-point” response to each and every comment is outlined below.

Comment 1:

Thank you for sending me the research article paper “ Regulation of long non-coding RNAs by plant secondary 2 metabolites: A novel anticancer therapeutic approach” for review in the Cancers. In the article of Kalhori et al., the author discussed the role of LncRNA and phytochemical in cancer prevention and treatment. Topic is interesting and novel. However, there are important points that should be improved.

Response:

We thank the reviewer for their expertise, time, and effort for reviewing our manuscript. We are deeply encouraged by the reviewer’s generous comments about the quality of our manuscript. We have addressed the reviewer’s specific comments as indicated below.

Comment 2:

Author should divide the heading into sub-heading like; 2. Methodology for literature search and selection, 3. lncRNAs: categories and functions and others. It would be helpful for the readers to read and understand.

Response:

We sincerely appreciate the reviewer’s comments. We have introduced several subsections to present information in section 3 (page 4, line 142 to page 5, line 187).

Comment 3:

4. Role of lncRNA in human cancers: this heading should be divided into several sub-headings. could be divided according to LncRNA or cancer types.

Response:

We are in absolute agreement with the reviewer. We have divided section 4 into the following subsections:

4. Role of lncRNAs in human cancers (page 5, lines 190-209)

4.1. Role in EMT (page 8, lines 219-237)

4.2. Role in NF-κB Signaling and Telomerase Activity (page 9, lines 238-265).

4.3. Role in Energy Metabolism (page 9, lines 266-285).

4.4. Role in Drug Resistance (page 10, lines 289-319).

Additional text has been added to various subsections as appropriate.

Comment 4:

Author should construct a graphical representation of the following heading: Mechanism of action of phytochemicals in the regulation of lncRNAs in cancer.

Response:

We are indebted to the review for this suggestion. Accordingly, we have introduced a new figure (Figure 3) to summarize the key points on mechanisms of action of phytochemicals in the regulation of lncRNAs in cancer  (page 24).

Additionally,

  1. The reference list has been modified as we have added several new references. Special attention is given to conform to the order of references and bibliographic style of the journal.
  2. The entire manuscript has been thoroughly checked and edited to ensure uniform style, organization, and quality.

On behalf of my co-authors, I once again express my sincere thanks to the erudite reviewer for the valuable suggestions and constructive input to improve the quality of our manuscript.